# Protective efficacy of holed and aging PBO-pyrethroid synergist-treated nets on malaria infection prevalence in north-western Tanzania

Eliud Lukole[1]*, Jackie Cook[2], Jacklin F. Mosha[1], Louisa A. Messenger[3], Mark Rowland[3], Immo Kleinschmidt[2,4,5], Jacques D. Charlwood[2], Franklin W. Mosha[6], Alphaxard Manjurano[1], Alexandra Wright[2], Natacha Protopopoff[2]

1 Department of Parasitology, National Institute for Medical Research, Mwanza Medical Research Centre, Mwanza, Tanzania, 2 MRC International Statistics and Epidemiology Group, Department of Infectious Disease Epidemiology, London School of Hygiene and Tropical Medicine, London, United Kingdom, 3 Department of Disease Control, London School of Hygiene and Tropical Medicine, London, United Kingdom, 4 Wits Research Institute for Malaria, School of Pathology, Faculty of Health Sciences, University of the Witwatersrand, Johannesburg, South Africa, 5 Southern African Development Community Malaria Elimination Eight Secretariat, Windhoek, Namibia, 6 Department of Parasitology, Kilimanjaro Christian Medical University College, Moshi, Tanzania

* ellylufi@ymail.com, Eliud.Lukole@lshtm.ac.uk

**Data Availability Statement:** Data in Tanzania are governed by strict law and cannot be shared outside the country without a Data Transfer

## Abstract

Two billion pyrethroid long-lasting insecticidal nets (LLINs) have been distributed since 2004 for malaria prevention in Sub-Saharan Africa. Current malaria control strategies rely on an assumed effective 3-year lifespan for LLINs. PBO synergist LLINs are a newly recommended class of net but there is limited information on their life span and long-term protective efficacy in communities. To assess their operational survival, a cohort of 390 PBO LLINs (Olyset Plus) and 367 standard pyrethroid LLIN (Olyset net) from 396 households were followed for 36 months in Western Tanzania. To assess the association between the condition of the LLIN and malaria infection, nets from at least 480 randomly selected households were assessed during malaria prevalence cross-sectional surveys at 4, 9, 16, 21, 28, and 33 months post-distribution. Information on the presence and condition of nets, and demographic information from the household, were collected to evaluate factors influencing net durability. After 3 years less than 17% of nets distributed still remained in the households. The fabric condition was not associated with malaria infection in either type of net. The difference between the net types was highest when nets were between 1–2 years old, when PBO nets appeared to be similarly protective as nets less than a year old, whereas standard nets were considerably less protective as they aged, regardless of fabric condition. There was no statistical difference in the estimated median functional survival time between net types with 1.6 years (95% CI 1.38–1.87) for PBO LLIN and 1.9 years (95% CI 1.67–2.06) for standard LLINs. After 3 years, there was a loss of 55% of permethrin (pyrethroid) content for both nets, and 97% of PBO content was lost in PBO LLIN. These results highlight that functional survival is less than the recommended 3 years for both net types. However,

agreement (DTA.pdf (nimr.or.tz)) and appropriate ethical approval. There was no provision under this study to share the data widely outside and was not mentioned in the consent form. A DTA process will need to be completed by interested researchers.

**Funding:** The study was funded under The Joint Global Health Trial scheme by UK Department for International Development, Medical Research Council, and Wellcome Trust and awarded to MR. The funders and intervention manufacturers had no role in study design, data collection and analysis, decision to publish, or preparation of the manuscript. The corresponding author had full access to all the data in the study and had final responsibility for the decision to submit for publication.

**Competing interests:** The authors have declared that no competing interests exist.

even as the nets age, the PBO nets remained more protective than standard nets, regardless of their condition.

## Background

Long-lasting insecticidal nets (LLINs) remain a cornerstone approach for malaria prevention [1] and 2 billion have been distributed between 2004 and 2021 in Sub Saharan Africa [2, 3]. The functional survival of LLINs is recommended to be at least 3 years under field conditions [4, 5]. Functional survival is characterised by the LLIN remaining in the household, in an acceptable physical condition, and retaining biological activity (giving protection against mosquitoes). Early accumulation of holes and loss of adequate insecticide is listed among the factors hindering LLINs' functional survival and protective effects [6, 7]. For decades, LLINs have been treated with pyrethroid insecticides, due to their high insecticidal activity and low mammalian toxicity [8]. However, recent evidence suggests that insecticide resistance in mosquito vectors may have reduced the level of protection provided by pyrethroid-only LLINs [9, 10].

One alternative to pyrethroid-only LLINs is to treat nets with a combination of pyrethroid insecticides and a synergist, piperonyl butoxide (PBO) [11]. Piperonyl butoxide (PBO) is not designed to kill insects directly, but when mixed and applied with pyrethroids enhances their potency by inhibiting enzymes that normally act to detoxify insecticides in the insect [12], essentially rendering pyrethroids effective, even in pyrethroid-resistant insects [13]. PBO is designed to be effective against mosquitoes whose resistance is based on oxidative metabolism [10]. There are currently two brands of LLIN which incorporate PBO which have been assessed in epidemiological trials: Permanet 3.0 [14] and Olyset Plus [10, 15]. In these trials, the PBO nets showed better efficacy than standard pyrethroid nets for at least 12 months of monitoring [10, 14, 15]. However, there are mounting concerns that the efficacy of PBO LLINs may wane before 3 years because the PBO has been shown to degrade at a fast pace [16]. A recent study conducted in Uganda showed that in nets that had been in use in the community, mosquito mortality measured using three-minute WHO cone bioassay decreased quickly however, still remained higher for PBO LLIN compared to standard LLIN over two years [17].

Net durability is a key factor in the effectiveness of nets. As nets deteriorate there is likely to be more vector-human contact; and, in addition, owners are more likely to stop using, discard, or repurpose holed LLINs as they perceive them to be no longer useful [18, 19]. To investigate the durability of PBO-nets, we performed a study to investigate the attrition (net loss), physical integrity (number of holes present), chemical content (amount of active ingredient remaining), and the association between holed and aged LLINs and malaria infection in children under 15 years, over three years of use in field conditions. The study took place within a 4 arm factorial cluster randomised control trial (RCT) to evaluate the efficacy of PBO LLIN compared to standard LLINs [20].

## Methodology

### Study area and study arms

The study was conducted in Muleba district ($1^0$ 45' S $31^0$ 40' E), in Kagera, in the North-West region of Tanzania on the Western shore of Lake Victoria. Muleba covers an area of approximately 3500km$^2$ at an altitude ranging from 1,100m-1,600m above sea level. The district comprises 43 wards and 160 villages and a population of 540,310 [21]. Rainfall occurs in two seasons: the short rains in October–December (average monthly rainfall 160 mm) and the

long rains in March-May (average monthly rainfall 300 mm). Malaria transmission in the district occurs throughout the year with two distinct peaks, in June to July and November to January following the long and short rains, respectively. The study area is described in detail elsewhere [10]. Briefly, it comprised 48 clusters from 40 villages within 13 wards. A total of 30,000 households were included. The 48 clusters were randomly assigned to one of four arms: (1) conventional standard LLIN (pyrethroid only), (2) PBO LLIN (pyrethroid and PBO), (3) conventional standard LLIN + Indoor Residual Spraying (IRS, with pirimiphos-methyl CS at the dosage of 1–2 g AI/m$^2$ [22]), and (4) PBO LLIN + IRS (Fig 1). Indoor residual spraying (IRS) took place once in February/March 2015.

**Net treatment and distribution.** As part of the trial, approximately 90,000 LLINs (45,000 PBO LLIN and 45,000 standard LLINs) were distributed to study clusters between 6$^{th}$-8$^{th}$ February 2015. The nets appeared similar and were blue, rectangular LLINs made from 150 denier polyethylene material. PBO LLIN (Sumitomo Chemicals, Japan) contains the pyrethroid

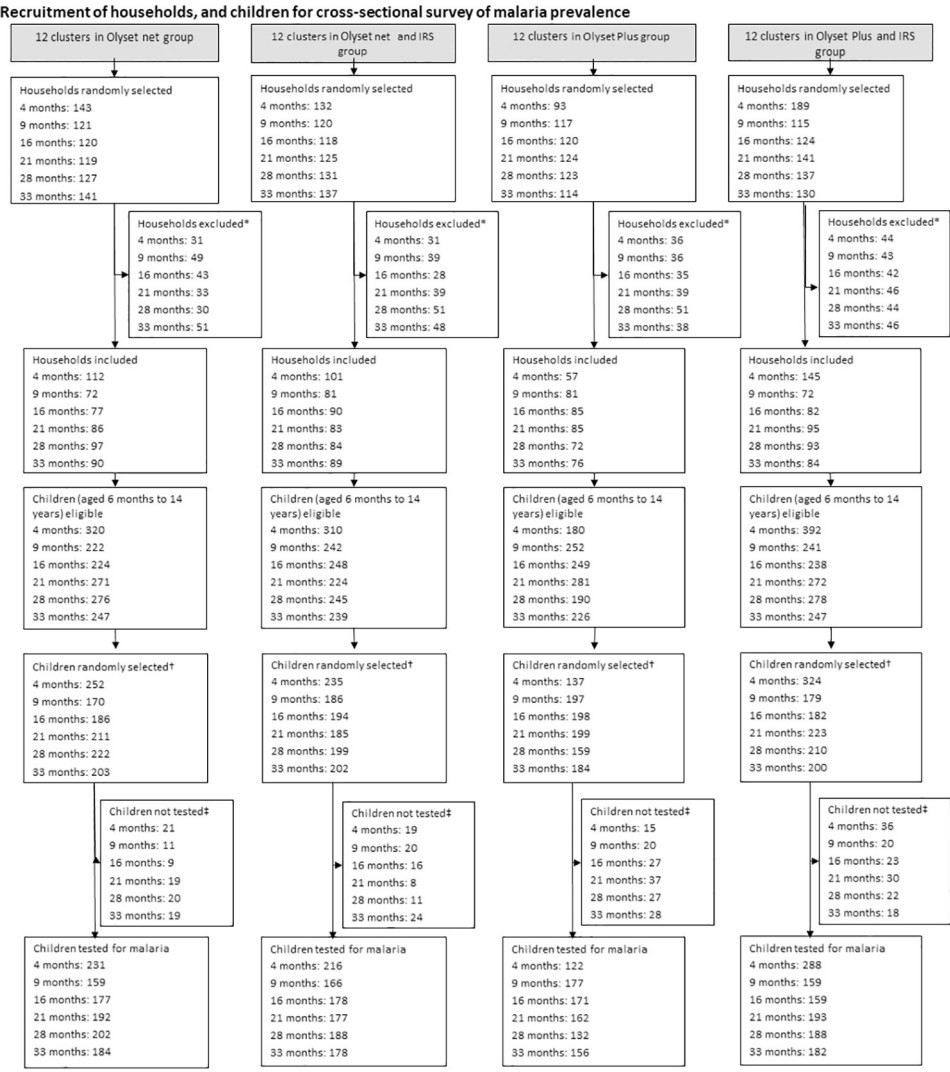

**Fig 1. Durability study profile.**

permethrin (20 g/kg) as the active ingredient (AI) and the synergist Piperonyl Butoxide (PBO) (10 g/kg) [23]. The standard LLIN (Sumitomo Chemicals, Japan) contains permethrin (20 g/kg) incorporated in the yarn but no synergist [24].

## Study design

This is a secondary analysis of the clinical data from the trial registered with ClinicalTrials.gov (NCT02288637).

**Longitudinal LLIN study.** The durability of the nets was assessed first in a longitudinal survey, which followed a cohort of nets every year for 36 months. For this study, 100 households per cluster were randomly selected from two clusters receiving PBO LLIN (one with and one without IRS) and two receiving standard LLINs (one with and one without IRS).

At enrolment, all nets in selected households hung and in use received a unique ID number, to be easily identified at subsequent visits. At each 12, 24, and 36 months visit, the physical presence of the net was recorded (to assess survivorship) and nets were inspected for holes to estimate functional survival of the LLIN [4]. In addition, 10 PBO LLIN and 10 standard LLINs still in use were randomly sampled from the households for insecticide content analysis and if the selected nets were missing, alternative nets in the same household were sampled. All sampled nets were replaced with new nets of the same brand but were not subsequently followed up for fabric integrity or chemical analysis.

**Cross-sectional LLIN study.** Net durability was also assessed in 480 households (10 households per cluster) during successive cross-sectional surveys done at 4, 9, 16, 22, 28, and 33 months post-distribution (Table 1).

At each time point, a random sample of 10 households per cluster was selected for LLIN physical durability assessment. If the selected household had no study net, households were not replaced by another. In each household, field assistants randomly selected three study LLINs. In the same selected household, up to three children aged 6 months to 14 years were randomly selected to be tested for malaria infection using a rapid diagnostic test (CareStart Malaria HRP2/pLDH(pf/PAN) Combo, DiaSys, UK). Children diagnosed as malaria positive by the rapid diagnostic test were treated with artemether-lumefantrine according to national guidelines. At the 21-month cross-sectional survey, 10 PBO LLIN and 10 standard LLINs were randomly selected to be used for insecticide content analysis.

**LLIN fabric integrity assessment.** To assess fabric integrity, nets were mounted onto a 170cm x 180cm x160cm collapsible frame to facilitate the visual assessment of the net [25]. The number of holes, location on the net, type of holes, and hole size were recorded. Size was classified into four categories as per WHO guidelines [4, 5]: size 1 = 0.5–1.99 cm, size 2 = 2–9.99 cm, size 3 = 10–25 cm and size 4 > 25 cm. Size of holes was measured by superimposing transparent plastic with illustrations of hole sizes.

**LLIN insecticidal content.** For each net collected for chemical content, one 30 cm x 30 cm net piece was cut from each side of the net following standard WHO procedure [5]. These pieces were then uniquely labelled, packed in aluminium foil, and stored at 4˚C; and sent to the Liverpool School of Tropical Medicine (United Kingdom) for chemical content analysis using High-Performance Liquid Chromatography (HPLC) [26].

## Data analysis

Data collection was done on Personal Digital Assistants (PDAs) using Pendragon Software. The primary outcome was LLIN functional survival (LLIN present and in serviceable condition). Secondary outcomes included (1) malaria infection in children sleeping under LLINs of different physical condition, and (2) chemical content. All statistical analyses were conducted

**Table 1. Measurement, frequency, and outcomes of durability components.**

| Component | Definition | Data collection survey | Measurement | Outcome indicators | WHO criteria | Timing of measures | Clusters included |
|---|---|---|---|---|---|---|---|
| Attrition [4,5,27] | Net no longer available in household due to discarding, alternative uses, given away, used elsewhere or stolen | Longitudinal survey: **Attrition rate-category1** | **Household follow-up survey**: all LLINs labelled and lost due to wear and tear | **Numerator**: Total number of each LLIN product no longer present in surveyed households due to wear and tear x 100 | | 12, 24, 36 months | 4 |
| | | Longitudinal survey: **Attrition rate-category2** | **Household follow-up survey**: all LLINs labelled and lost because they are given away, stolen, sold, used in another location, or withdrawn for chemical content analysis | **Numerator**: Total number of each LLIN product no longer present in surveyed households because they are given away, stolen, sold, used in another location, or withdrawn for chemical content analysis in surveyed households x 100 | | | |
| | | Longitudinal survey: **Attrition rate-category3** | **Household follow-up survey**: all LLINs labelled and lost because are being used for other purposes | **Numerator**: Total number of each LLIN product no longer present in surveyed households that are being used for other purposes in surveyed households x 100 | | | |
| | | | | **Denominator**: Total number of each LLIN product distributed to surveyed households | | | |
| Physical integrity [4,5,27] | Physical state of the net | Longitudinal survey | Number, location, and size of hole(s) for all labelled nets in HH | Holed surface area (HS) | Good: HS < 80cm$^2$, Damaged: HS 80-789cm$^2$, extremely torn: HS $\geq$ 790cm$^2$ | 12, 24, 36 months | 4 |
| | | Cross-sectional survey | Number, location, and size of hole(s) for 3 nets per HH | | | 4, 9, 16, 21, 28, 33 months | 48 |
| Functional survival [4,5,27] | Estimation of nets still in households and in good and damaged (serviceable) condition | Longitudinal survey | Median survival analysis | Total LLINs present and serviceable / (Total LLINs originally labelled at baseline—total net given away or not followed at each time point) | Median net survival in years = time point at which the estimate of functional survival crosses 50% | 12, 24, 36 months | 4 |
| Holed nets protective effect | Ability of each torn LLIN product to provide protection against malaria infection | Cross-sectional survey | Children between 6 months and 14 years tested for malaria parasite by RDT | Odds ratio of malaria infection between users of holed standard LLIN and PBO LLIN in children between 6 month and 14 years, over three years of use in field conditions | Good: HS <80cm$^2$, Damaged: HS 80-789cm$^2$, extremely torn: HS $\geq$ 790cm$^2$ | 4, 9, 16, 21, 28, 33 months | 48 |
| Insecticide content | Amount of active ingredient (PBO and/or pyrethroid) per gram of the LLIN as determined by chemical assay | Cross-sectional survey | Permethrin and PBO content in g/kg in 10 pieces (30x30 cm pieces) per net | Concentration at baseline, and over study period as per WHO | Permethrin: 20 g/kg, PBO: 10g/kg | 21 months | 48 |
| | | Longitudinal survey | | | | 0, 12, 24, 36 months | 4 |

using STATA release 15 (StataCorp, College Station, TX, USA). Household social and economic wealth indices were constructed and analysed by Principal component analysis (PCA), and households were sub-divided into wealth tertiles. The *svy* command using cluster as sampling unit was used to account for the clustered design of the study. Comparisons between the two types of nets were tested for significance using the Chi-square ($\chi^2$) test allowing for the clustered design.

A sum of the holes identified in the nets were weighted [27] to calculate a hole surface area (HS) = (1.23 x no. of size1 holes) + (28.28 x no. of size2 holes) + (240.56 x no. of size3 holes) + (706.95 x no. size4 holes) [27]. Linear regression allowing for survey design was used to compare hole surface area between net products, using log-transformed data to normalise the distribution. Based on the hole surface area, LLINs were assigned to different categories (good $<80cm^2$; damaged = 80–789 $cm^2$, and extremely torn $\geq$790 $cm^2$) to determine risk factors associated with the physical state of each LLIN product.

Survivorship, attrition and functional survival were estimated using the longitudinal data and compared between net types and over the three visits using Chi-square ($\chi^2$), accounting for the clustered design. Median lifetime of nets was estimated using Kaplan-Meier estimators and a hazard ratio for difference in functional survival was calculated using Cox regression, adjusting for the clustered nature of the survey design. To aid in functional survival calculations, a questionnaire was used to capture information on net condition and usage, as well as "how many months ago the net was lost/discarded" if the net was no longer in the household at each follow-up point. Then, the number of months was subtracted from the current date to determine the actual month the net was lost. For the longitudinal data, where nets were sampled at multiple timepoints, generalized estimation equations (GEE) using a logit model were used to determine factors associated with the poor physical condition of the 2 net brands accounting for the repeated measures on individual nets and correlations within clusters. In addition, we assessed the factors which were associated with bad net condition using multivariable logistic regression with the cross-sectional data. Factors investigated included net type, net age, socio-economic status (SES) of household, hanging habits of nets, type of beds, type of mattress, net ever washed, indoor residual spraying (IRS), and use of open flames in the household. The association between net physical condition and malaria infection was assessed using multivariable logistic regression, controlling for SES, child age, indoor residual spraying (IRS), and presence of eaves in the house.

## Ethics

The trial was approved by the ethics review committees of the Kilimanjaro Christian Medical University College, the London School of Hygiene & Tropical Medicine, and the Tanzanian Medical Research Coordinating Committee (NIMR/HQ/R.8a/VolIX/1803). A trial steering committee reviewed progress. Written informed consent from parents or guardians was obtained for each survey.

## Results

### Nets included in the study

In total, 757 nets (397 PBO LLIN, 360 standard LLIN) from 396 households from four clusters were included in the longitudinal study and followed up for 3 years at 12 months intervals. The proportion of households that were lost to follow-up was 21% (3%-refused, 14%-dwelling vacant for survey duration, and 4% dwelling not found) over the 3 years of the trial. Cross-sectional surveys were conducted every year in June-July and November-December from 2015 to 2017. In total, 2104 LLINs (987 PBO LLIN, and 1117 standard LLINs) from 1,383 households were assessed during cross-sectional surveys. During cross-sectional surveys, the proportion of households with at least one study net for every two people decreased from 50% after 4 months to 7% after 33 months. The proportion of participants reporting using study nets the night before decreased from 71% at 4 months to 21% at 33 months post-intervention.

The demographics of households over surveys and study arms were relatively similar in terms of the average number of sleeping rooms, number of sleeping spaces, education status,

**Table 2. Household and socioeconomic characteristics of participating households in longitudinal and cross-sectional surveys.**

| Covariates | Longitudinal survey | Cross-sectional survey | | | | | |
|---|---|---|---|---|---|---|---|
| Net Age | | 4 months | 9 months | 16 months | 21 months | 28 months | 33 months |
| Mean number rooms (sd), N | 4.7 (1.5), 350 | 5.1 (1.5), 405 | 5.0 (1.3), 298 | 5.2 (1.6), 326 | 5.4 (1.8), 340 | 5.5 (2.1), 339 | 5.0 (1.6), 331 |
| Mean number sleeping spaces (sd), N | 2.3 (1.0), 350 | 2.2 (0.8), 405 | 2.3 (0.8), 298 | 2.3 (0.9), 326 | 2.4 (0.9), 340 | 2.4 (0.9), 339 | 2.4 (0.8), 331 |
| Mean number of people per household (sd), N | 5.1 (2.5), 396 | 5.7 (1.9), 405 | 5.7 (1.8), 298 | 5.6 (2.3), 326 | 5.6 (1.9), 340 | 5.6 (1.9), 339 | 5.5 (1.8), 331 |
| % Malaria infection (in children 0.5–14 years) (N), 95%CI | | 40 (948), 34–47 | 32 (732), 27–37 | 38 (760), 30–45 | 49 (818), 40–57 | 72 (790), 65–78 | 50 (789), 42–57 |
| % literate heads of households (N), 95%CI | 74 (351), 66–80 | 75 (405), 71–79 | 77 (298), 72–81 | 78 (326), 73–82 | 74 (340), 69–78 | 75 (339), 71–80 | 80 (331), 76–85 |
| **Proportion of households with at least one net for every two people (HH Access) % (N)** | | | | | | | |
| Any LLIN | | 89 (460) | 87 (349) | 74 (371) | 65 (447) | 61 (441) | 66 (407) |
| Study PBO LLIN (Olyset Plus) | | 51 (219) | 46 (167) | 26 (179) | 18 (232) | 10 (203) | 8 (204) |
| Study standard LLIN (Olyset net) | | 49 (241) | 40 (182) | 22 (192) | 13 (215) | 12 (238) | 6 (203) |
| **Proportion of participants reporting using a net the night before % (N)** | | | | | | | |
| Any LLIN | | 76 (2324) | 76 (1749) | 49 (1884) | 53 (1968) | 59 (1913) | 48 (1879) |
| Study PBO LLIN (Olyset Plus) | | 72 (1134) | 70 (894) | 44 (926) | 43 (1026) | 38 (885) | 23 (887) |
| Study standard LLIN (Olyset net) | | 69 (1190) | 71 (855) | 44 (958) | 39 (942) | 44 (1028) | 20 (992) |
| **Household possessions** | | | | | | | |
| % electricity (N), 95%CI | 8 (351), 3–18 | 5(405), 3–8 | 5 (298), 3–10 | 8 (326), 5–13 | 9 (340), 6–14 | 12 (339), 8–18 | 19 (331), 14–25 |
| % mobile phones (N), 95%CI | 70 (351), 58–81 | 69 (405), 64–74 | 69 (298), 61–75 | 71 (326), 65–76 | 73 (340), 67–78 | 69 (339), 62–76 | 79 (331), 75–84 |
| % open eaves: (N), 95%CI | 69 (351), 60–77 | 59 (405), 54–64 | 61 (298), 54–68 | 67 (326), 61–73 | 63 (340), 56–70 | 46 (339), 40–53 | 47 (331), 40–54 |
| **Main housing materials** | | | | | | | |
| % floor: earth/sand (N), 95%CI | 88 (349), 74–95 | 89 (405), 83–92 | 96 (298), 92–98 | 91 (326), 88–94 | 89 (340), 84–93 | 85 (339), 79–89 | 83 (331), 76–88 |
| % roof: tin (N), 95%CI | 86 (350), 77–91 | 82 (405), 76–87 | 83 (298), 78–87 | 91 (326), 86–94 | 92 (340), 87–95 | 93 (339), 90–95 | 92 (331), 87–95 |
| % walls: mud (N), 95%CI | 61 (350), 27–87 | 78 (405), 71–84 | 79 (298), 72–85 | 73 (326), 66–80 | 76 (340), 69–83 | 72 (339), 63–89 | 65 (331), 57–72 |

sd: standard deviation; CI: confidence intervals; N: total number of observations; LLIN: Long-lasting insecticidal net

main housing materials (Table 2). There was an increase in the number of households with electricity over the study period, from 5% to 19% (Table 2). This was due to the government's mission to facilitate access to modern energy services in rural Mainland Tanzania through Rural Energy Agency (REA).

## LLIN survivorship and attrition

Three years after net distribution, survivorship (defined as nets present in surveyed households and available for sleeping under) was 28.8% (95% CI 11.8–54.9) for PBO LLIN and 25.6% (95% CI 22.2–29.3) for standard LLIN. A small proportion of nets were given away, stolen, sold, or used in other locations at each time point and these also did not differ significantly per arm (Table 3). Attrition due to wear and tear (poor net condition) increased from 3% (95% CI 0.3–23.6) at 12 months to 56% (95% CI 37.6–72.4) at 36 months in PBO LLIN and from 4% (95% CI 3.5–4.5) to 63% (95% CI 57.1–68.7) in standard LLINs (Table 3). There were no significant differences in attrition between PBO LLIN and standard LLINs over the study period (p = 0.6816). Comparing survivorship between arms which received IRS and those that did

**Table 3. Survivorship and attrition of cohort nets by net type and age (data from longitudinal survey).**

| LLIN type | Standard LLIN (Olyset net) (N = 360) | | | PBO LLIN (Olyset Plus) (N = 397) | | |
|---|---|---|---|---|---|---|
| Net age | 12 months | 24 months | 36 months | 12 months | 24 months | 36 months |
| Nets labelled in visited households | N = 329 | N = 308 | N = 309 | N = 326 | N = 313 | N = 309 |
| Attrition rate-category1: % (95% CI) | 4.0 (3.5–4.5) | 34.4 (30.0–39.1) | 63.1 (57.1–68.7) | 3.1 (0.3–23.6) | 31.9 (11.6–62.7) | 55.7 (37.6–72.4) |
| Attrition rate-category2: % (95% CI) | 6.1 (4.4–8.4) | 10.1 (4.8–20.0) | 11.3 (9.3–13.8) | 6.7 (4.7–9.7) | 14.4 (6.7–28.1) | 15.5 (11.6–20.5) |
| Attrition rate-category3: % (95% CI) | 0.3 (0.0–3.7) | 1.0 (0.4–2.2) | 0 (0–0) | 0.3 (0.0–4.5) | 0.6 (0.6–0.7) | 0 (0–0) |
| Missing nets* (n) | 31 | 52 | 51 | 71 | 84 | 88 |

Attrition rate-category1: for nets that have been destroyed or disposed of due to wear and tear (poor condition) in surveyed households; Attrition rate-category2: for nets not available for sleeping under for reasons other than poor fabric integrity (given away, stolen, sold or used in another location, withdrawn by PAMVERC staff) in surveyed households; Attrition rate-category3: for nets used for other purposes in surveyed households;

* Missing nets from households that were not interviewed due to either (dwelling vacant, a dwelling not found, and refused) and were not included in the denominator when calculating survivorship and attrition

not, revealed a significant difference in survivorship at 24 months (p = 0.044) and 36 months (p = 0.050); where arms without IRS recorded higher survivorship of nets (S1 Table). There was no significant difference in functional survival at 3 years (defined as the presence of serviceable LLIN) of the 2 net products; 15.4% (95% CI 4.8–39.7) for PBO LLIN and 17.9% (95% CI 14.7–21.5) for standard LLIN (p = 0.6929). Estimated median functional survival was (1.9 years (95% CI 1.67–2.06) for standard LLIN and 1.6 years (95% CI 1.38–1.87)) for the PBO LLIN (S2A and S2B Table). There was statistical difference in median lifetime between nets (regardless of net type) in IRS and non-IRS arms, with lower median lifetime in IRS arms (p = 0.0103) (S4 Table).

## Net physical integrity

After 3 years of use, on average, 93% of standard LLIN and 94% PBO LLIN had at least one hole of any size in the longitudinal survey; while on average, 63% of standard LLIN and 57% of PBO LLIN had at least one hole in the cross-sectional data (Table 4). The damage to PBO LLIN was more severe than to standard LLINs across the study period in the longitudinal survey but differences were only significant at 36 months. Based on WHO categories, 55% of PBO LLIN and 37% of standard LLINs were considered extremely torn after 3 years of field use in the longitudinal survey; whereas in the cross-sectional survey data it was 24% of PBO LLIN and 33% standard LLINs at 33 months post-distribution. Of the nets present in the households but no longer in use as they were perceived unprotective by users after 3 years, 32% were in good or damaged condition (7%-intact, 5%-good, and 20%-damaged). In both surveys, the of the lower half of the fabric of both net types were more vulnerable to tear than other parts of the net (S3 Table).

## Factors associated with net physical integrity

Factors that could be associated with net fabric integrity were assessed. These included net type, net age, SES, type of bed, type of mattress, folding/unfolding the net in the morning, and washing of the net (Table 5). The influence of these factors to fabric integrity differed by survey (longitudinal and cross-sectional survey). In the cross-sectional data, factors that were significantly associated with increased net damage included net age, lowest SES, unfolding the net in the morning, and sleeping under grass or reed mats and washing of the net. In the longitudinal data; net age, unfolding the net in the morning, and sleeping without a bed frame were significantly associated with increased net damage.

**Table 4. LLIN physical durability expressed by hole surface area by survey (cross-sectional and longitudinal) and net age.**

| Net type | Standard LLIN (Olyset Net) | | | | PBO LLIN (Olyset Plus) | | | | p-values for the comparison of mean hole surface area between PBO LLIN and Standard LLIN |
|---|---|---|---|---|---|---|---|---|---|
| | Total Net assessed | % net with at least 1 hole | % extremely torn LLIN (HS $\geq$790cm$^2$) | Hole surface area (cm$^2$): Median (IQR) | Total Net assessed | % net with at least 1 hole | % extremely torn LLIN (HS $\geq$790cm$^2$) | Hole surface area (cm$^2$): Median (IQR) | |
| **Longitudinal survey** | | | | | | | | | |
| 12 months | 296 | 91 | 19 | 89 (6–577) | 294 | 87 | 29 | 234 (5–1116) | 0.5763 |
| 24 months | 171 | 93 | 41 | 368 (39–1840) | 168 | 98 | 49 | 770 (92–2284) | 0.1015 |
| 36 months | 79 | 95 | 37 | 322 (31–1520) | 89 | 97 | 55 | 925 (212–2464) | 0.0462 |
| **Cross sectional survey** | | | | | | | | | |
| 4 months | 289 | 39 | 4 | 31 (4–138) | 273 | 24 | 2 | 30 (2–490) | 0.9328 |
| 9 months | 224 | 42 | 13 | 271 (58–994) | 182 | 46 | 14 | 180 (28–1247) | 0.3982 |
| 16 months | 188 | 67 | 24 | 270 (60–1311) | 162 | 65 | 26 | 402 (28–1421) | 0.7701 |
| 21 months | 150 | 70 | 26 | 375 (87–1326) | 170 | 63 | 22 | 321 (25–1077) | 0.1257 |
| 28 months | 178 | 78 | 22 | 398 (111–911) | 120 | 73 | 24 | 196 (33–1349) | 0.0703 |
| 33 months | 88 | 82 | 33 | 376 (65–1488) | 80 | 71 | 24 | 357 (79–1464) | 0.8657 |

* net considered as extremely torn if hole surface area $\geq$790 cm$^2$; CI: confidence interval

## Chemical content

At baseline (0 months), all 10 PBO LLIN and 10 standard LLIN samples complied with their target doses of pyrethroid (20 ± 5 g AI/kg) and PBO concentration (10 ± 2.5 g PBO/kg) for PBO LLIN nets (Fig 2). At 12 months, 24 months, and 36 months mean permethrin content in PBO LLIN decreased to 15 g/kg, 6 g/kg, and 9 g/kg, corresponding to a loss of 29.8%, 73.4%, and 56.9% of the original dose, respectively. The mean permethrin content of standard LLINs was 21 g/kg at 12 months, 16 g/kg (23.8% loss of the original dose) at 24 months, and 10 g/kg (53.3% loss) at 36 months. Mean PBO content decreased to 3 g/kg (69.5%) at 12 months and less than 1g/kg (96% loss) after 36 months. Pyrethroid content of nets collected at 21 months during the cross-sectional survey was 12.2 g/kg for PBO LLIN and 16.7 g/kg for standard LLIN and PBO concentration in PBO LLIN was 1.4 g/kg.

## Malaria infection and physical condition of LLIN

Malaria infection was measured in cross-sectional surveys only. Between year two and three, there was a significant increase in malaria infection in all arms. When controlling for other factors, the fabric condition was not associated with malaria infection in either type of net. However superior protection afforded by PBO nets depended on the age of nets: with the PBO LLIN providing particularly strong protection versus standard nets when both net types were aged between 1 and 2 years (Table 6). For nets that were still relatively new (<1 year), there was only weak evidence that PBO nets were better in protecting against infection; after 2 years

**Table 5. Factors associated with LLIN being classed as unserviceable (HS: $\geq$790cm$^2$) (data from longitudinal and cross-sectional survey).**

| Covariate | Longitudinal survey[a] | | | | Cross-sectional survey[b] | | | |
|---|---|---|---|---|---|---|---|---|
| | %extremely torn (HS $\geq$ 790cm$^2$) (N) | aOR | 95% CI | p-value | %extremely torn (HS $\geq$ 790cm$^2$) (N) | aOR | 95% CI | p-value |
| **Net type** | | | | | | | | |
| Olyset net | 28.2 (546) | 1 (Ref) | | | 17.5 (1117) | 1 (Ref) | | |
| Olyset plus | 39.2 (551) | 1.77 | 1.23–2.57 | *<0.001* | 16.1 (987) | 0.81 | 0.60–1.09 | 0.160 |
| **Net age (years)** | | | | | | | | |
| Year 1 | 23.7 (590) | 1 (Ref) | | | 7.6 (968) | 1 (Ref) | | |
| Year 2 | 44.8 (339) | 2.41 | 1.56–3.71 | *<0.001* | 24.5 (670) | 3.60 | 2.60–4.97 | *<0.001* |
| Year 3 | 46.4 (168) | 1.68 | 0.95–2.97 | 0.080 | 24.9 (466) | 3.62 | 2.50–5.24 | *<0.001* |
| **Socio economic status (SES)** | | | | | | | | |
| Lowest | 38.4 (315) | 1 (Ref) | | | 20.5 (639) | 1 (Ref) | | |
| Medium | 36.9 (393) | 1.12 | 0.72–1.73 | 0.620 | 15.6 (716) | 0.72 | 0.52–0.99 | 0.040 |
| Highest | 26.7 (389) | 0.76 | 0.46–1.25 | 0.270 | 14.8 (749) | 0.86 | 0.62–1.19 | 0.370 |
| **How was the net found** | | | | | | | | |
| Hanging loose over sleeping space | 35.3 (309) | 1 (Ref) | | | 20.7 (781) | 1 (Ref) | | |
| Hanging folded | 21.6 (393) | 0.56 | 0.39–0.79 | *<0.001* | 14.3 (1216) | 0.75 | 0.58–0.97 | 0.030 |
| **Type of bed** | | | | | | | | |
| No bed frame | 35.5 (248) | 1 (Ref) | | | 18.8 (640) | 1 (Ref) | | |
| Stick | 27.4 (237) | 0.65 | 0.42–1.00 | *0.050* | 18.7 (646) | 1.17 | 0.85–1.60 | 0.330 |
| Wood/Iron | 22 (287) | 0.56 | 0.34–0.91 | *0.020* | 13.7 (736) | 0.88 | 0.62–1.24 | 0.476 |
| **Type of mattress** | | | | | | | | |
| Grass/reed mat/clothes | 33.5 (269) | 1 (Ref) | | | 22.9 (637) | 1 (Ref) | | |
| Foam/spring mattress | 25.1 (502) | 0.91 | 0.59–1.38 | 0.65 | 14.2 (1385) | 0.69 | 0.51–0.92 | 0.010 |
| **Net ever washed** | | | | | | | | |
| No | 32.6 (656) | 1 (Ref) | | | 12.9 (1441) | 1 (Ref) | | |
| Yes | 37.8 (415) | 0.89 | 0.60–1.34 | 0.59 | 25.1 (630) | 1.40 | 1.06–1.85 | 0.020 |
| **Indoor residual spraying (IRS)** | | | | | | | | |
| No | 30.7 (583) | 1 (Ref) | | | 18.5 (1026) | 1 (Ref) | | |
| Yes | 37.2 (514) | 1.44 | 1.00–2.07 | 0.05 | 15.2 (1078) | 0.88 | 0.66–1.18 | 0.400 |

[a] analysis was done using logistic generalized estimation equations (GEE);

[b] analysis was done using multivariable logistic analysis;

aOR: adjusted odds for all variable in the table; Ratios; HS: Hole surface area

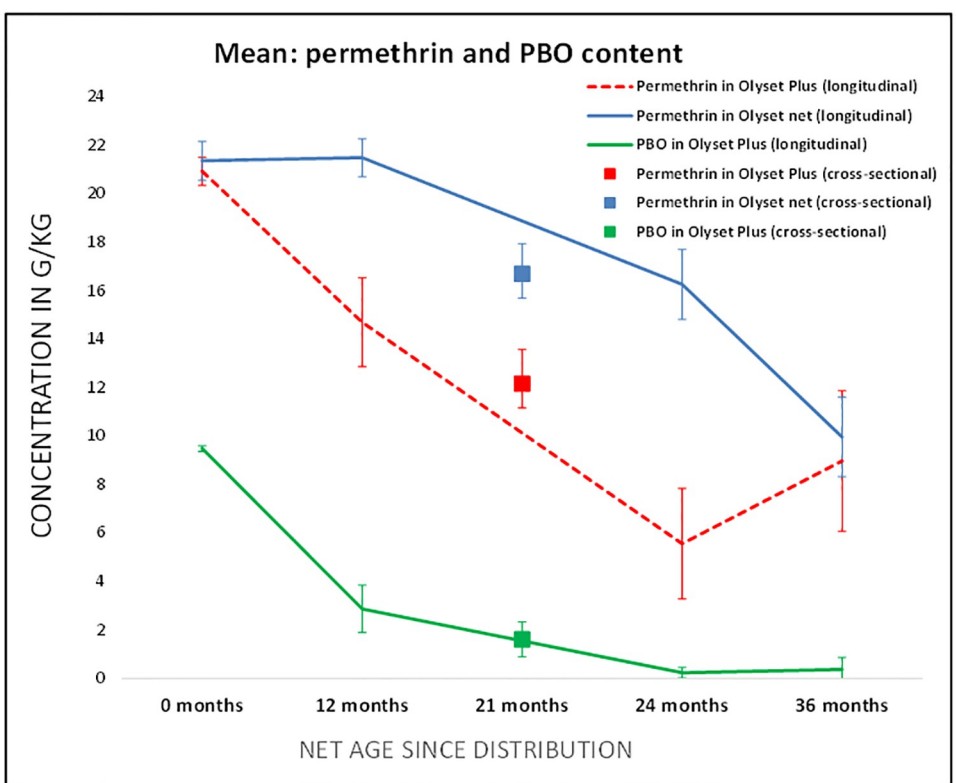

Longitudinal nets to assess chemical content were collected at 0, 12, 24 and 36 months post net distribution. Cross-sectional nets were collected once at 21 months post net distribution

**Fig 2. Insecticide content over 3 years of field use.**

there was also little evidence that PBO nets provided enhanced protection compared to standard nets.

## Discussion

This study presents the durability of two types of nets with and without the synergist PBO over 3 years of field use and examines whether this was associated with malaria infection in an area

**Table 6. Multivariable adjusted odds ratios of malaria infection among 0.5–14-year olds using nets with different physical conditions.**

| Covariates | Malaria Prevalence % (N) | Adjusted odd Ratios* | 95% CI | p-value |
|---|---|---|---|---|
| **Net types per condition of net** | | | | |
| Olyset plus vs Olyset net: Less than 1 year | 32.8 (540) vs 39 (564) | 0.80 | 0.46–1.38 | 0.4197 |
| Olyset plus vs Olyset net: 1 year to < 2 years | 32.7 (312) vs 50.3 (322) | 0.44 | 0.26–0.75 | *0.0028* |
| Olyset plus vs Olyset net: 2 years to < 3 years | 57.1 (191) vs 64.6 (243) | 0.88 | 0.49–1.60 | 0.6791 |
| **Net condition** | | | | |
| Good (HS: < 80 cm²) | 39.5 (1174) | 1 (Ref) | | |
| Damaged (HS: 80–789 cm²) | 45.9 (342) | 0.99 | 0.71–1.36 | 0.9330 |
| Extremely torn (HS: ≥ 790cm) | 47.0 (387) | 1.10 | 0.79–1.51 | 0.5790 |

*adjusting for survey rounds, socio-economic status (SES), children age, indoor residual spraying (IRS), and presence of eaves in the house; There was evidence of interaction between net age and net type, p value = 0.0392; PBO LLIN (Olyset plus) vs Standard LLIN (Olyset net)

of pyrethroid resistance. As has been seen with other studies [18, 28], the nets in this study did not remain in serviceable condition for three years which is required for 'long-lasting' nets, both in terms of fabric integrity and chemical content.

The majority of nets followed in the longitudinal study had at least one hole by 12 months with poor fabric integrity stated as the main reason for attrition. Nets collected via the cross-sectional surveys were generally less damaged, this is likely to be due to the fact that participants in the longitudinal study were asked to keep hold of their nets, whilst it appears that if this was not the case, users are more likely to get rid of the torn nets quickly. In addition, we do not have data on when the nets sampled cross-sectionally were first used, and some may have been stored for several months before being put into use, as has been seen elsewhere [29, 30]. This highlights the differential information the two methods (longitudinal and cross-sectional) obtain with regards to damage to fabric integrity over time, compared to the point when users realistically discard their nets. Indeed, a third of the nets that were considered unprotective by users which were examined by the field workers were actually still in serviceable condition per WHO guidelines. Community attitudes towards damaged nets may vary from place to place and may feed into differences in durability reported between countries [28].

Overall, PBO nets were more likely to be torn, although this was only evident in longitudinal survey nets and not in the cross-sectional nets. The higher survivorship (presence of nets in the household) despite fewer being classed as in serviceable condition per WHO guidelines of PBO nets by 12 months, suggests that users held onto them even if they were in bad condition. This could be due to the higher protective efficacy of the PBO nets, regardless of fabric condition [31].

Both net types lasted for less than 2 years, which is a shorter timespan than has been seen in other studies' for similar nets [16, 18, 32]. Moreover, in this study, less than a third of the distributed nets remained in households after 3 years of community use, which is slightly lower than has been observed in other studies conducted in Tanzania (45%) [18], Mozambique (40%) [33], and Kenya (82%) [34]. Notably, a recent study in Kenya examining the durability of the same make of PBO LLIN showed extremely high survivorship of 91% after 3 years of use [16]. Differences in living conditions but also in net care behaviour could explain these variations [35]. Noticeably, net survivorship in the arms that did not receive indoor residual spraying (IRS) was significantly higher than in arms that received IRS, suggesting that people are more likely to care for their nets if there is no other preventive intervention in place. This is unlikely to be a chance finding since communities were randomly allocated to receiving IRS, and may therefore have implications for any future strategy that combines different interventions. A similar effect may have been seen in clusters where non-study nets were readily available to replace lightly damaged study nets- the durability of the nets therefore may be underestimated in settings where other interventions are abundant.

Several risk factors were identified for increased risk of damage to the net. Washing of the net and aging of the net increased the likelihood of net damage whereas folding the net every morning was observed to protect the nets against damage, as has been seen in other studies [33, 36, 37]. Other factors that could protect the net against damage included sleeping under bed frames and the use of foam mattresses. Targeted behavioural change communication (BCC) campaigns could increase awareness on the net care, repair and net durability determinants thereby preventing LLINs from becoming extremely torn [36–39].

In this study, fabric condition was not associated with malaria infection in either type of net. However, older nets were associated with higher risk of infection, with differential effects by net type depending on age of net. Notably, the difference between the net types was highest when nets were between 1–2 years old, when PBO nets appeared to be similarly protective as nets less than a year old, whereas standard nets were considerably less protective as they aged,

regardless of fabric condition (Table 6). This suggests that a major benefit of PBO nets may be in mitigating the detrimental effect of aging of nets, but that this advantage relative to standard nets wanes after two years. Therefore, to protect the increased efficacy that PBO nets offer in combating malaria, programmes and funding agencies should consider adapting net distribution strategies to the effective net lifespan or alternatively physical and chemical durability of the nets need to be improved to meet the three years replacement cycle. Currently, it seems clear that swathes of the community are using considerably less effective nets for at least a year before the next net distribution cycle [40].

The results also suggest that much of the protection given by the nets is related to the insecticide (which declined in both nets over the three years), rather than the condition of the fabric [41]. Other studies have also found that malaria infection is not necessarily associated with the condition of the nets [7, 42], however, we would caution against interpreting this too strongly as study net usage considerably declined over the three years of this study and low community-level coverage of nets could be confounding the impact of fabric quality on malaria infection.

Both net types had chemical content within the manufacturers' specifications at the start of the study. However, PBO content was low at 12 month and almost gone by 24 months. The active ingredient (permethrin) was considerably reduced in both nets, but reduced more quickly in PBO nets. Despite insecticide loss and aging of nets, PBO nets were more protective against malaria infection in the second year than standard LLINs, suggesting that people should continue using PBO LLINs even if they are aged and damaged [10]. This may also explain why there was a high proportion of extremely torn PBO LLIN compared to standard LLINs in use at the end of the study; as users may still find that they are effective, despite the physical condition and aging. These findings concur with the earlier findings on torn PBO LLIN, that they will continue protecting against blood-feeding *Anopheles* even if they are damaged or torn [31].

Access and usage of study nets decreased sharply over the study period likely due to the high attrition rate leading to too few study nets remaining in the households at the end of the study. However, the higher access and usage of non-study nets suggests that residents are keen to keep using nets if they are considered in good condition. In general, the nets which replaced study nets were obtained from periodic governmental net distributions in the study area. It is of key importance that manufacturers produce more durable nets, which could potentially be achieved with a small increase in unit price [18, 43] by reinforcing seam and net lower midzone or bottom part which is vulnerable to abrasion.

The study had some limitations. The low study net usage by the third year may have confounded some of the results, though it also highlights the speed at which these nets are discarded in the community. In addition, the insecticide content was tested on smaller number of nets than what is recommended by WHO, although, a study that assessed insecticide content of PBO nets on a number of nets 4 times higher than of this study, found a relatively similar reduction in chemical content [16]. Importantly, this study does not report on entomological bio-efficacy meaning that we do not know whether the reduced chemical content on the nets over time would have also resulted in reduced mosquito mortality. Finally, due to higher attrition than expected, textile integrity could only be assessed on a small sample of nets after 3 years. However, the high rate of attrition in this community is clear and highlights that the usable life of these two nets is well under the recommended lifespan of 3 years.

## Conclusion

The findings of this study demonstrate that the nets' lifespan in this setting was below 2 years. Aged PBO LLINs still provided better protection against malaria infection than standard

LLINs up to 2 years. It is important that users are aware of the benefits of using nets, even if they become torn- and targeted behaviour change communication campaigns may help to reduce the attrition of torn nets. Moving forward, to ensure population coverage with effective nets, either manufacturers need to make nets that last for the recommended 3 years, in terms of fabric and chemical content, or distribution methods need to be increased in regularity to ensure that the population is able to sleep under effective bed nets.

## Supporting information

**S1 Table. Survivorship and attrition of cohort nets by age and arms with and without Indoor Residual Spraying (IRS) (data from longitudinal survey).**
(DOCX)

**S2 Table.** A. Proportion of net still present in the house and in serviceable condition (= functional survival) by net product and time point and median survival in years. B. Cox regression analysis presenting unadjusted and adjusted Hazard ratio.
(DOCX)

**S3 Table. Geometric mean of holes by zone and by size.**
(DOCX)

**S4 Table. M0065dian survival of the nets by study arms.**
(DOCX)

**S1 Fig. Kaplan Meier plots for survival life of nets.**
(TIF)

## Acknowledgments

We thank colleagues and staff at the Kilimanjaro Christian Medical University College in Muleba and Moshi, and those at the National Institute of Medical Research in Mwanza who were involved in the project. We acknowledge the assistance provided by staff at the Muleba District Medical Office, at both the ward, village and hamlet level. Additionally, we thank USAID/President's Malaria Initiative and RTI International for funding and implementing the indoor residual spraying operation in the study area, and the Tanzania Communication and Development Center for communications and implementation of the distribution campaign of the long-lasting insecticidal nets. We thank Hanafy Ismail (Liverpool School of Tropical Medicine, Liverpool, UK) for chemical analysis of the long-lasting insecticidal net. Finally, we thank all the participating households.

## Author Contributions

**Conceptualization:** Eliud Lukole, Jackie Cook, Natacha Protopopoff.

**Data curation:** Eliud Lukole, Jacklin F. Mosha, Alexandra Wright, Natacha Protopopoff.

**Formal analysis:** Eliud Lukole, Jackie Cook, Immo Kleinschmidt, Natacha Protopopoff.

**Funding acquisition:** Mark Rowland, Natacha Protopopoff.

**Investigation:** Natacha Protopopoff.

**Methodology:** Eliud Lukole, Natacha Protopopoff.

**Project administration:** Eliud Lukole, Alexandra Wright, Natacha Protopopoff.

**Software:** Eliud Lukole, Natacha Protopopoff.

**Supervision:** Eliud Lukole, Jacklin F. Mosha, Alexandra Wright, Natacha Protopopoff.

**Visualization:** Natacha Protopopoff.

**Writing – original draft:** Eliud Lukole.

**Writing – review & editing:** Eliud Lukole, Jackie Cook, Jacklin F. Mosha, Louisa A. Messenger, Mark Rowland, Immo Kleinschmidt, Jacques D. Charlwood, Franklin W. Mosha, Alphaxard Manjurano, Alexandra Wright, Natacha Protopopoff.

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
