## [Decision Letter · Decision Letter 0]

28 Jun 2022

PGPH-D-22-00608

Protective efficacy of holed and aging PBO-pyrethroid synergist-treated nets on malaria infection prevalence in north-western Tanzania

Dear Dr. Lukole,

Thank you for submitting your manuscript to PLOS Global Public Health. After careful consideration, we feel that it has merit but does not fully meet PLOS Global Public Health’s publication criteria as it currently stands. Therefore, we invite you to submit a revised version of the manuscript that addresses the points raised during the review process.

Please submit your revised manuscript by . If you will need more time than this to complete your revisions, please reply to this message or contact the journal office at globalpubhealth@plos.org. Please include the following items when submitting your revised manuscript:

We look forward to receiving your revised manuscript.

Kind regards,

Bernt Lindtjorn, PhD

Academic Editor

Journal Requirements:

1. Thank you for indicating that the study reported within the manuscript text is a secondary analysis of the clinical trial registered: NCT02288637. For the purposed of transparency please clearly indicate this within the manuscript text. Furthermore it is also not necessary to provide the study protocol as supporting information, we recommend that this document is removed.

2 .Please amend your detailed Financial Disclosure statement. This is published with the article. It must therefore be completed in full sentences and contain the exact wording you wish to be published.

State the initials, alongside each funding source, of each author to receive each grant.

3. Please update your online Competing Interests statement. If you have no competing interests to declare, please state: “The authors have declared that no competing interests exist.”

4. In the online submission form, you indicated that “Deidentified data will be made available upon reasonable request to the corresponding author. There is no legal or ethical restrictions to the data.”. All PLOS journals now require all data underlying the findings described in their manuscript to be freely available to other researchers, either 1. In a public repository, 2. Within the manuscript itself, or 3. Uploaded as supplementary information.

5. Please provide separate figure files in .tif or .eps format and ensure that all files are under our size limit of 10MB.

Additional Editor Comments (if provided):

Reviewers' comments:

Reviewer's Responses to Questions

**Comments to the Author**

1. Does this manuscript meet PLOS Global Public Health’s publication criteria? Is the manuscript technically sound, and do the data support the conclusions? The manuscript must describe methodologically and ethically rigorous research with conclusions that are appropriately drawn based on the data presented.

Reviewer #1: Partly

Reviewer #2: Yes

2. Has the statistical analysis been performed appropriately and rigorously?

Reviewer #1: Yes

Reviewer #2: Yes

3. Have the authors made all data underlying the findings in their manuscript fully available (please refer to the Data Availability Statement at the start of the manuscript PDF file)?

Reviewer #1: Yes

Reviewer #2: Yes

4. Is the manuscript presented in an intelligible fashion and written in standard English?

Reviewer #1: Yes

Reviewer #2: Yes

5. Review Comments to the Author

Reviewer #1: This manuscript presents the analysis of a dataset generated on net longevity and effectiveness from a clinical trial conducted in Tanzania. The manuscript covers a generally well conducted and important study and provides important information for malaria vector control programs and researchers. Its valuable and presents important details to researchers, though some of the effectiveness information has been previously discussed in other reports. Nevertheless, this study presents significant added value to those earlier reports on the main trial findings.

There are a number of minor points and few major issues that should be addressed before this manuscript is fully ready for publication.

Major issues:

1) The authors present findings from net longevity in the context of a factorial RCT. The authors present findings on net longevity and also note that in two of the four trial arms, the ITNs were combined with IRS. They note that having IRS alongside of the ITNs significantly decreased net longevity. This finding is very important for the overall study and may be one of the more important findings of the entire paper as it has significant policy implications for the use of combination interventions, which the authors note. Despite this, It seems that they present unadjusted net longevity estimates, (at least I assume) averaging across the nets where IRS was present and where IRS was not present. While these are likely statistically accurate representations of the nets in the trial, doing so greatly reduces the generalizability of their findings. IRS is deployed in a relatively limited scope, is generally not recommended to be done in combination with ITNs by WHO, and therefore, the estimates of net lifetimes without IRS would be much more helpful for the general malaria community to make use of these findings (the authors could at least make the focus showing their adjusted results so that readers could see both).

2) The corollary to the above is that there appear to have been significant numbers of non-study nets available in the study area. The authors recognize that participants may have substituted IRS coverage for ITNs leading to a lower durability in the IRS arm. But they do not discuss or acknowledge that similar phenomenon may have occurred with availability of non-study nets as well. The availability of new non-study nets, might also affect lifetimes of nets in a similar way, leading to earlier disposal of less damaged nets when replacements are available.

3) Lastly there are a couple of statements in the conclusion that are not well founded in the evidence presented in the manuscript.

4) Since one of the main outcomes of the study is median lifetime, it would be very useful to see the kaplan meir plots for lifetimes of nets, rather than only the plot of of the insecticide decay chart.

More minor concerns:

First paragraph of study design: This is mildly confusing, the heading is longitudinal LLIN study, but the paragraph describes two studies, a cohort study and a repeated cross sectional studies. I would suggest that the former is a longitiduinal study and the latter is a cross sectional study. The authors could do a better job making these two studies/data collection modalities stand out.

last line of this paragraph, nets chosen for sampling which had already been "attritted" were replaced with existing nets. This is fine as a study design, but it leads to an overestimate of how much insecticide and or intact fabric remains over time, because it is an estimate of how much insecticide and intact fabric are there on nets that remain in use in the original house. THe authors need to be careful to frame all these results in this context.

Data analysis, svy command used to adjust SEs for clustering at RCT clusters. Were there more than one net per household in study? If so was SE inflation for within household clustering considered? IT seems that there would be significant within household correlation based on the association of SES and net lifetime alone.

Functional survival over time was calculated using cox regression? Usually this is done with a KM estimator? COX regression would be used to estimate hazard ratios associated with determinants of hazard.

The use of multivariate as a term for a regression generally implies multiple outcomes. IT is probably not the right term for the logistic regressions here multiple logistic or multivariable logistic regression is probably more accurate. Especially if this logistic regression is not specified using a variance components model, as seems to be the case here where the svy: prefix was used in STATA rather than a Random effects estimator.

Net physical integrity section: There is a error printout for a missing reference in the manuscript.

last sentence of this section, sentence should not start with however. This sentence is difficult to understand and should be re-written.

Chemical content of nets: Careful to frame these results in the context of this is only assessed among nets which remain in use and identifiable and as such is probably an overestimate of the remaining insecticide that was initially distributed. (i.e. this analysis has a survivor bias built in).

First paragraph of discussion... missing references.

IN general: the discussion contains some sloppy wording. The authors should be careful to use the term "MEDIAN Lifetime" or something specific which relates to their analysis rather than simply saying "the nets don't last three years" throughout. This lack of precision leads to divorcing their findings form their conclusions and a lack of clarity around how big this discrepancy actually is.

First sentence of second discussion paragraph: This is very hard to follow and probably a run on sentence, should be rewritten.

"examined by the field workers were actually still in serviceable condition. Community

attitudes towards damaged nets may vary from place to place and may feed into differences in

durability seen across countries(30)." - please say were in serviceable condition per WHO guidelines or something, they were not considered serviceable by the households.

"PBO nets had less functional survival by 12

months, suggesting that users had held onto them even if they were in bad condition" - Ifound this sentence very hard to parse. Would suggest rephrasing.

"This is not a chance finding since

communities were randomly allocated to receiving IRS, and may therefore have implications for

any future strategy that combines different interventions." - RCTs can still result in findings by chance. But I agree that this is likely due to the effect of having IRS in those areas, that indeed it isnt due to chance. It also reinforces the idea that subsitute goods such as IRS or new nets might serve to reduce the lifetime of nets. The authors report this for IRS but neglect it as a threat to generalizability of findings....

"Therefore, to maximize the gain that PBO nets offer in combating malaria,

programmes and funding agencies should consider a two-year cycle for net replacement

strategies. " - THIS Statement should go out, it does not follow scientifically or logically from their findings. The relative gains from deploying PBO nets compared to non-PBO nets of a similar age are indeed greater in the second year, but this tells us nothing about what cycle a campaign would follow to maximize an undefined "gains." Leaving aside that maximizing gains without some accompanying constraint such as a budget ceiling is not a sensible goal since it leads to the logical conclusion of a one year cycle or a three week cycle being even better than a two year cycle, the idea that using a two year cycle rather than a three or four or one year cycle would maximize gains, relies only on the relative gains of arm A vs. B here. e.g. a two year cycle of PBO distribution, compared to a two year cycle of standard net distribution would in this case create the maximum difference in gains between the two alternatives. This is not the same thing as maximum gains (no matter how you describe them).

"These

findings concur with the earlier findings on torn PBO LLIN, that they will continue protecting

against blood-feeding Anopheles irrespective of physical condition(31)." - It strains credulity that this is "Irrespective" of physcial condition.... but agree taht damaged nets could still protect. Suggest rewording.

CONCLUSION:

There is no evidence provided in this study about whether BCC might actually affect net lifetimes. Suggest dropping this statement entirely.

"Importantly, either manufacturers need to make

nets that last for the recommended 3 years, in terms of fabric and chemical content, or

distribution methods need to be increased in regularity to ensure that the population is able to

sleep under effective bed nets"

Neither of these statements can be concluded based on data and findings presented here. In fact don't you actually report that the shorter lived nets had a greater impact in this study? I recognize that this is confounded by the inclusion of PBO in those nets, but from this data and the way it was analyzed, it seems net life is actually inversely related to impact.

I'd suggest instead that you stick closer to the data, and assert that these are not meeting a three year median lifetime in this setting. ITs not clear at all what making a longer lasting net will do to overall impact from the data and analysis you present here. Additionally increasing campaign frequency to make up for net lifetime has not been analyzed here, so i would refrain from drawing conclusions about it. This would result in a dramatic increase in cost, it may itself have the effect of shortening net life times. I think its fine to clearly state that the additional advantage of PBO nets relative to regular nets seem to be less meaningful or not existent after two years, which suggest the possibility of greater impact with a shorter campaign cycle, but to make that recommendation without actually analyzing it seems a step too far.

ery nets

Reviewer #2: This paper answers an important question and was clearly written.

I have only minor comments:

Methods:

I would rename the section called "Study area" since it contains information on the arms of the trial this study is embedded in, not just the study area.

Table 1: The authors could add a column on how many clusters each component was measured in.

Table 1: Is survivorship just 1 - attrition? if so there's no need for both. If not, a better explanation of the difference should be given.

The section on the "longitudinal LLIN study" has information about the cross-sectional study, which is confusing because there is a separate section on this. It would be easier to keep the information on each study in its own section.

In the section on "cross-sectional LLIN study" could the authors specify the time points so avoid confusion with the longitudinal study.

Results:

Table 3: The 3 attrition different rates should be explained in table 1.

table 3: the "attrition rates" are not actually rates (rates have a denominator of time) so please just say attrition.

6. PLOS authors have the option to publish the peer review history of their article (what does this mean?). If published, this will include your full peer review and any attached files.

**Do you want your identity to be public for this peer review?** For information about this choice, including consent withdrawal, please see our Privacy Policy.

Reviewer #1: No

Reviewer #2: No

---

## [Editor Report · Decision Letter 1]

22 Sep 2022

Protective efficacy of holed and aging PBO-pyrethroid synergist-treated nets on malaria infection prevalence in north-western Tanzania

PGPH-D-22-00608R1

Dear Mr Lukole,

We are pleased to inform you that your manuscript 'Protective efficacy of holed and aging PBO-pyrethroid synergist-treated nets on malaria infection prevalence in north-western Tanzania' has been provisionally accepted for publication in PLOS Global Public Health.

The journal requires the reader to have access to anonymized data sets. This is required by the rules of PLOS Global Public Health and must be done before this publication can be accepted.

Best regards,

Bernt Lindtjorn, PhD

Academic Editor
